# In-situ transfer vat photopolymerization for transparent microfluidic device fabrication

Yang Xu [1,2], Fangjie Qi[1,2], Huachao Mao[1,3], Songwei Li[1,4], Yizhen Zhu[1,4], Jingwen Gong[1,5], Lu Wang [5], Noah Malmstadt[5,6] & Yong Chen [1,2,4✉]

While vat photopolymerization has many advantages over soft lithography in fabricating microfluidic devices, including efficiency and shape complexity, it has difficulty achieving well-controlled micrometer-sized (smaller than 100 μm) channels in the layer building direction. The considerable light penetration depth of transparent resin leads to over-curing that inevitably cures the residual resin inside flow channels, causing clogs. In this paper, a 3D printing process — in-situ transfer vat photopolymerization is reported to solve this critical over-curing issue in fabricating microfluidic devices. We demonstrate microchannels with high Z-resolution (within 10 μm level) and high accuracy (within 2 μm level) using a general method with no requirements on liquid resins such as reduced transparency nor leads to a reduced fabrication speed. Compared with all other vat photopolymerization-based techniques specialized for microfluidic channel fabrication, our universal approach is compatible with commonly used 405 nm light sources and commercial photocurable resins. The process has been verified by multifunctional devices, including 3D serpentine microfluidic channels, microfluidic valves, and particle sorting devices. This work solves a critical barrier in 3D printing microfluidic channels using the high-speed vat photopolymerization process and broadens the material options. It also significantly advances vat photopolymerization's use in applications requiring small gaps with high accuracy in the Z-direction.

[1] Center for Advanced Manufacturing, University of Southern California, Los Angeles, CA 90007, USA. [2] Daniel J. Epstein Department of Industrial and Systems Engineering, University of Southern California, Los Angeles, CA 90089, USA. [3] School of Engineering Technology, Purdue University, West Lafayette, IN 47907, USA. [4] Department of Aerospace and Mechanical Engineering, University of Southern California, Los Angeles, CA 90089, USA. [5] Mork Family Department of Chemical Engineering and Materials Science, University of Southern California, Los Angeles, CA 90089, USA. [6] Department of Chemistry, University of Southern California, Los Angeles, CA 90089, USA. ✉email: yongchen@usc.edu

A microfluidic chip consists of intrinsically connected microchannels or chambers within a bulk material with inlets and outlets. The liquid fluid is directed, mixed, or split by microchannels' network to achieve the desired applications in chemistry and biomedical fields such as microreactors[1], fluid mixer[2–4], cell analysis/culture[5–8], and cell/particle sorting[9–11]. Typically, for truly microfluidic devices, at least one channel dimension should be in the range of 10–100 μm (which is also the size range of cells)[12]. And channel transparency is always desired for easy visualization in this application domain (e.g., to observe fluorescently labeled samples).

3D printing as a promising method to fabricate microfluidic chips has attracted considerable attention. Many researchers reported that 3D printing is superior to conventional PDMS micro-molding in fabricating microfluidic chips with increased structural complexity and dramatically reduced fabrication time and cost by circumventing mold-making and labor-intensive procedures[13–16]. Vat photopolymerization (VPP), such as stereolithography, is the most promising 3D printing technology for microfluidic chip fabrication because of its high resolution, smooth surface quality, and affordability[16,17]. In the VPP process, an irradiation light source such as a scanning laser beam or a projected image pattern is used to cure the vat's liquid resin[18,19]. Standard connectors such as barbed connectors for fluid and pressure sources can be integrated into the VPP-printed microdevices to achieve plug-and-play. Besides, flushing unpolymerized resin in microvoids after the VPP process is much easier than removing solid sacrificial support required by other 3D printing processes such as material jetting and material extrusion[20–22]. Also, most microfluidic channels are self-supported by liquid resin in VPP. In comparison, although self-supporting structures for material extrusion have been proposed before based on the maximum printable overhang angle[1,23,24], the accordingly fabricated microfluidic channels had cross-sections that were in the large microfluidic regime (100–500 μm) or at the sub-100 μm level at best (Supplementary Table 1 and Fig. 1a). More importantly, the imposed cross-sectional profile constraints not only complicate the design phase but also limit its use in applications that require precise laminar flow control or in structures with low aspect ratios, such as perfusion systems with concentration gradient control[25,26], single-cell trapping devices[27–29], and particle sorting devices[9]. For material jetting, a technique of carefully laying thin polycarbonate membranes onto open channels during fabrication was developed; hence the embedded closed channels were converted into open channels with no need for support material[30]. Nevertheless, like soft lithography, this membrane-sealing approach also faces manual alignment and bonding issues, imposing limitations in 3D fluidic networks.

Despite the advantages over material extrusion and material jetting in fabricating microfluidic chips, current VPP technologies also face a severe limitation in the Z-axis resolution that is critical for microfluidic devices. For the channel height dimension, most VPP-based 3D printers can only fabricate channels with >200 μm height when printing transparent resin required for microfluidic channels (Supplementary Table 1 and Fig. 1a). The practical limit of minimum channel height results from resin's over-curing[31–33]. That is, the light irradiation required to build the channel roof can potentially photopolymerize the resin residing inside the channel. In addition, additional light will penetrate the previously built layers when curing the subsequent layers due to resin transparency, leading to channels' blockage. Researchers have investigated different strategies to solve this issue, which can be divided into three categories. One type of the strategies is to fine-tune the printing parameters[34,35]. The second type is to decrease the light penetration depth of the used liquid resin by adding photosensitizing additives for the visible blue light source

(405 nm)[35,36], or shifting the light source from visible blue light to ultraviolet (UV) light (≤385 nm)[37,38], or further adding UV absorbing dyes[8,39–41]. However, these two kinds of methods will either result in channels >100 μm or render the printed structures colored, bringing difficulty to observe the fluidic flow when using a microscope[41]. A more recent UV absorber was reported to solve the over-curing issue while avoiding colored parts[42]. The last type is to combine conventional manufacturing methods such as micro-molding for bulk device fabrication and nanoscale 3D printing techniques such as two-photon polymerization (TPP) to fabricate critical features[43]. However, significant efforts are required in switching between the two processes. Also, the maximum printable key feature size would be limited since the submicron curing size of TPP is poorly suited to fabricate large-size features such as channel roofs[43]. In general, for VPP, light penetration depth $\delta_p$ governs the Z-resolution of microfluidic chips. That is, the minimum feasible channel height for a given photocurable resin must be larger than 2.3 times its light penetration depth[39] (Fig. 1b and Supplementary Table 1). This rule dramatically decreases the material options available for 3D-printed microfluidic devices.

This paper presents a VPP process called In-situ Transfer VPP (IsT-VPP) to reliably produce microfluidic channels of 10 μm height without additional requirements on liquid resins such as reduced transparency (Fig. 1a). Compared with other VPP-based methods using customized light engines or modified photocurable resins, we used the most commonly used 405 nm light source and commercial transparent resin with a light penetration depth of 179.1 μm (Fig. 1c). As in Fig. 1b, our approach breaks the minimum channel height limit associated with the light penetration of the liquid resin; hence more material options can now be used for microfluidic chip fabrication. The key idea of IsT-VPP is to print the channel-roof layer separately via double exposure on the resin vat surface using an additional build platform (called aux platform in the paper). When printing the channel roof (i.e., the top layer portion that encloses the channel), the aux platform is utilized to prevent the light penetration into the residual liquid resin inside the channel. The channel roof is then in-situ transferred to the built part with the second exposure of a planned mask image. All the other layers are printed using the normal VPP process without the aux platform (Supplementary Fig. 4). The algorithm to generate corresponding mask images in the two exposures is also given (Supplementary Table 5). Our printing process's efficacy and versatility are demonstrated by fabricating multifunctional devices, including 3D microfluidic channels, microfluidic valves, and particle sorting devices. The results show that our approach is universal and can be applied to commercially available resins without any modifications. Current commercial and research VPP developers can easily apply our method to their systems. The widely available VPP machines and resins will significantly expand 3D printing in future microfluidic device fabrication.

## Results

**Light dose distribution.** Fabrication of microfluidic chips emphasizes specifically the microchannels inside the bulk. The resin trapped in the channels must remain liquid or gel state until the end of fabrication for it to be washed away. The curing of all the layers determines the total light dose of the channel portion. Therefore, the polymerization state of the liquid resin in the flow channels is governed by the following energy relationship[35,39].

$$D_t = \sum_{i=1}^{n} D_i(I_i, t_i, z = z_t) \le D_c \quad (1)$$

Where $D_t$ is the accumulated light dose delivered to the channel top $z_t$ (layer height starting from the build platform, as shown in

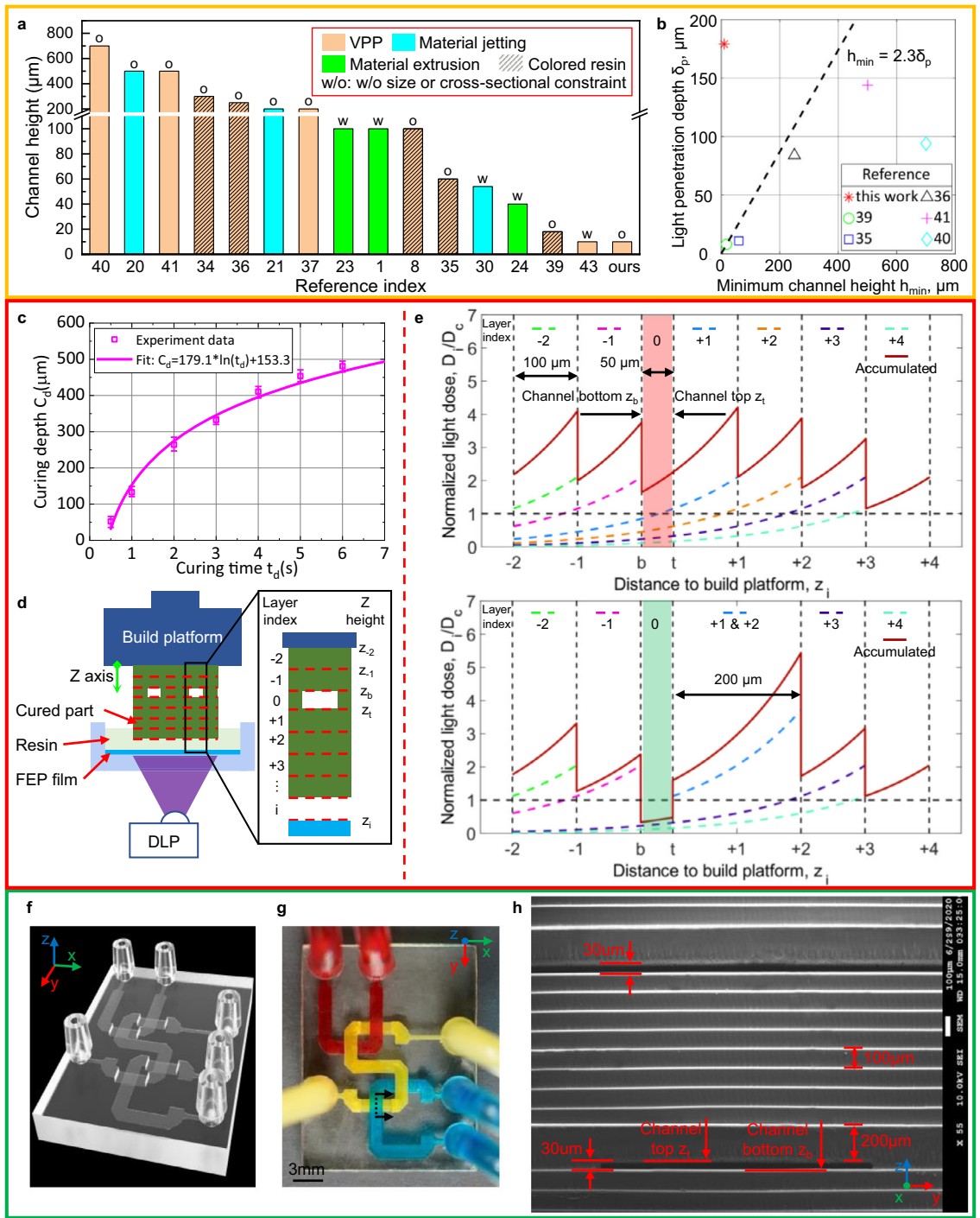

**Fig. 1 Illustration of the effect of resin optical property and light dose distribution on VPP-based microfluidic channels fabrication. a** Summarization of literature on minimum printable channel height $h_{min}$ and corresponding technical features. **b** Relationship between light penetration depth $\delta_p$ and minimum printable channel height $h_{min}$ for VPP-based processes. **c** Measured results and fitted curve of curing depth $c_d$ and exposure time $t_d$ for transparent resin. **d** Schematic diagram showing challenges in 3D printing transparent microfluidic channels. **e** Normalized light dose distribution of each projection and the accumulated light dose along the Z-direction when manufacturing the part in Fig. 1d via VPP (Top) and IsT-VPP (bottom). **f** CAD model of the crisscrossing USC-shaped fluid router. **g** Top view of the USC-shaped fluid router fabricated by IsT-VPP. **h** SEM image showing the cross-section of the channels indicated by the dashed line in Fig. 1g. The result demonstrates 30-μm-height channels.

Fig. 1d) when curing the subsequent $n$ layers. $D_C$ is the critical dose that the photocurable resin needs to solidify it; $D_i(I_i, t_i, z)$ represents the light dose distribution along the building direction when curing layer $i$ with the light intensity $I_t$ and exposure time $t_i$. Based on Beer-Lambert's law, the light energy falls off exponentially from the penetration surface. That is $D_i(I_i, t_i, z = z_t) = t_i I_i e^{-(z_i-z_t)/\delta_p}$,

where $\delta_p$ denotes light penetration depth characterizing the material optical property. According to Jacob's working curve $c_d = \delta_p \ln(\frac{t_d}{T_c})^{35}$, $\delta_p$ can be experimentally measured for a photocurable resin by fitting the paired values—curing depth $c_d$ and the corresponding exposure time $t_d$ (Fig. 1c and Supplementary Method I). $T_c$ denotes the critical exposure time, which is only

related to the liquid resin and the light intensity. From the fitted curve in Fig. 1c, the light penetration depth is 179.1 μm. The slope of the fitted curve indicates the sensitivity of the curing depth to the exposure time. The high sensitivity at smaller curing depths increases the depth control difficulty in printing microchannels. The error bars also indicate it is difficult to accurately control channel roof's thickness using only the energy-related methods.

In addition, excess light energy resulting from the subsequent layer exposures will cure the resin trapped inside the channels in the conventional VPP process (Fig. 1d). The top half of Fig. 1e gives the light dose distribution in the $Z$-direction corresponding to Fig. 1d. The curing depth $c_d$ is slightly higher than the layer thickness $l$ (e.g., $c_d = 1.25l$) to ensure sufficient bonding between layers. Layer 0 represents the channel area. $z_b$ represents the channel bottom, and $z_t$ specifies the channel top. If the accumulated light dose in the channel area ($z = z_b \sim z_t$) exceeds the threshold $D_c$ (i.e., the normalized light dose $D_t/D_c$ is >1), it will lead to occlusion of the channel (highlighted in red in the top half of Fig. 1e). Increasing channel roof thickness can barely mitigate light dose for transparent materials, which have large light penetration depths (blocked channel highlighted in red in Supplementary Fig. 1). Decreasing light penetration depth $\delta_p$ by making materials opaque (e.g., 20 μm, about 1/10 of that of transparent materials) can alleviate the problem; however, it still leads to a partial cure of the liquid resin inside the channel (partially blocked channel highlighted in red in Supplementary Fig. 2a), bringing inaccuracy and uncertainty. It is particularly challenging to precisely control the total light dose starting from the channel top by accurately controlling each layer's curing depth.

To address the VPP's overcuring challenge, a facile and general method of utilizing an aux platform (Fig. 2) to cure a thicker layer (e.g., 150-μm-height roof) for the layer adjacent to the channel is adopted. The aux platform blocks further transmission of light, thus lowers total energy absorbed by the resin inside the channel. The light energy caused by curing the after-roof layers is controlled to be smaller than the threshold $D_c$. The corresponding normalized light dose distribution of each projection and the accumulated light dose along the $Z$-direction is shown in the bottom half of Fig. 1e. In contrast, our IsT-VPP process significantly reduces $D_t$ lower than $D_c$ (channel area highlighted in green in the bottom half of Fig. 1e and Supplementary Fig. 2b). Hence the microfluidic channel can be fabricated successfully. A USC-shaped fluid router (Fig. 1f–h) was fabricated using IsT-VPP to demonstrate its high accuracy in the assembly free fabrication of 3D microfluidic devices with complex flow patterns. Fig. 1f gives the oblique view of the CAD model. Each letter is an individual microchannel (1400 μm × 30 μm) crisscrossing other channels in two different layers. Fig. 1g shows the top view of the 3D-printed result. The integrated fluid ports were designed as 1/16 in. barbed connectors. Opaque pigments (red, yellow, and blue) were used to flow through the channels for easy visualization. A partial cross-section of the channel indicated by the dashed line in Fig. 1g is presented in Fig. 1h using scanning electron microscopy (SEM). Both channel heights are 30 μm. And the inner channel surface is smooth and flat. A 1 mm spacing was designed between the two channels (the segments of the letter S and C) in the $Z$-direction. The details of the IsT-VPP process are discussed in the following sections.

**Experimental setup and process design**. To realize the above process for microfluidic channel fabrication, we created the In-situ Transfer Vat Photopolymerization Apparatus (IsT-VPPA), as shown in Fig. 2a. Our setup has an aux platform driven by two additional motorized linear stages compared with the traditional

bottom-up-projection-based VPP[44,45]. To ensure the cured layer will be separated from the aux platform in the in-situ transfer process, we coated the aux platform with PDMS and covered the resin vat with fluorinated ethylene propylene (FEP) film. We further control the contact areas of the cured layer with the two constraint surfaces, as explained later in this section. Note the aux platform will only be used for the microfluidic channels' roof layers by serving as a constrained surface, like the top-down-projection-based VPP[46]. A simplified $Y$-junction fluidic mixer model (Fig. 2c) is used to illustrate the printing process (Fig. 2d–i and Supplementary Movie 1).

(1) The bottom of the model, a $m$-mm thick cube, is 3D printed first using the normal VPP process on the main build platform. The corresponding mask image is shown in Fig. 2d.

(2) For the $h$-mm high channel, the top ($h$-$\varepsilon$)-mm thick portion is also fabricated by the main platform (Fig. 2e), where $\varepsilon$ is double exposure tolerance (e.g., 10 μm used in our study). Before finishing the residual $\varepsilon$-mm high channel, the channel-roof layer next to the flow channel will be fabricated in the following two steps.

(3) The aux platform will be used to print the channel roof portion to enclose the channel (Fig. 2f). Hence the main platform is lifted first to make room for the aux platform. The aux platform then moves leftward and downward to form an $l_r$-mm gap with the resin vat's FEP film ($l_r$ is an increased layer thickness to reduce the light penetration from building subsequent layers, the bottom half of Fig. 1e). A grayscale mask image is projected, also shown in Fig. 2f. The roof part's grayscale level is set to the maximum value of 255 to cure the roof thoroughly. The channel roof extends $\tau$-mm (e.g., 100 μm in our study) on both sides to be slightly wider than the channel width $w$. The aux platform constrains the top surface of the channel roof to ensure channel accuracy and surface quality. The rest portion of the layer will be semi-cured using a lower grayscale level (100 ≤ grayscale level ≤ 200). This will increase the roof feature's bonding force with the resin vat, which is crucial in our process and explained in the next step.

(4) When the aux platform moves back to the original idle position, the printed roof will remain stationary at the resin vat surface rather than following the aux platform's movement (Fig. 2g). Hence it is critical for the roof's bonding force with the resin vat to be larger than the one with the aux platform. We integrated three mechanisms to ensure this. First, the resin vat is coated with FEP film, while the aux platform is covered with PDMS. By investigating the bonding force of different material interfaces under different exposure areas (Supplementary Method II), a larger force is always needed to break the polymer-FEP film interface than the polymer-PDMS interface given the same contact area (Fig. 2b). Second, the contact area between the roof and the aux platform is smaller than the area between the roof and the resin vat using pre-planned mask images with grayscale values (Fig. 2f). This further increases the bonding force difference since the bonding force of a layer increases with its contact area with the coating media, while the channel area in a layer is usually orders of magnitude smaller than the area of the whole layer (Fig. 2b). Third, as Zhou[44] experimentally verified, the adhesive bonding strength between a cured layer and a coating media will increase with the input exposure energy. Hence the adhesion between the roof and the resin vat is naturally stronger since the roof's bottom

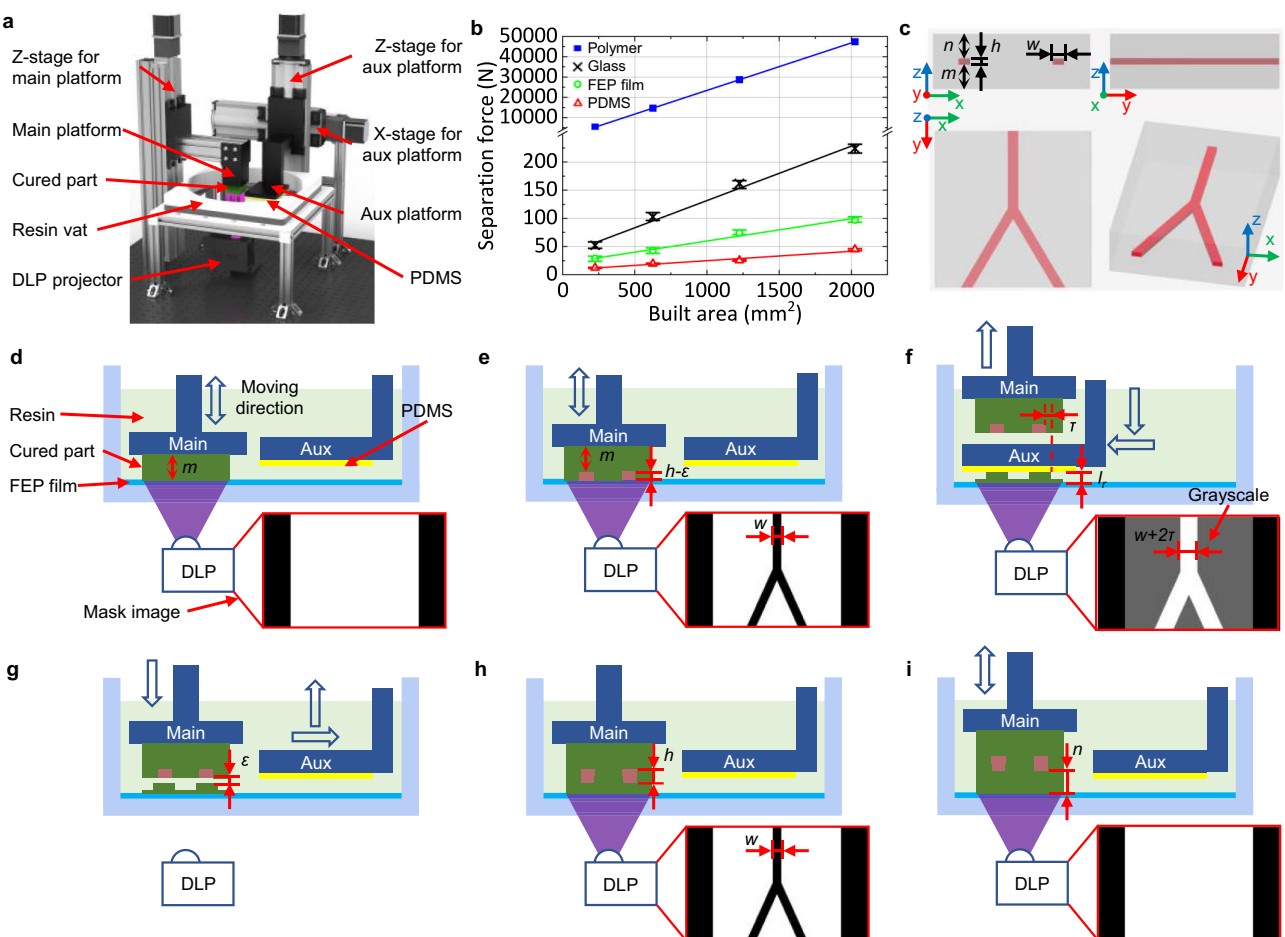

**Fig. 2 Principle of IsT-VPP process. a** The detailed structure of the IsT-VPP apparatus. **b** Measured separation forces of different contact areas for different material interfaces during VPP printing. Given the same contact area, the separation force between the part and the aux platform corresponding to the polymer-PDMS interface is the smallest. The separation force between the part and the vat surface represented by the polymer-FEP film interface is larger than the polymer-PDMS interface. A glass sheet works as the main build platform because the polymer-glass interface has stronger bonding than the previous two interfaces and is weaker than the polymer-polymer interface. The bonding force between neighboring layers of the printed part is derived from the resin's ultimate strength. **c** A simplified Y-junction fluidic mixer model is used to illustrate the IsT-VPP process and its three orthographic views. **d**, **e** Fabrication of the bottom portion and partial channel using only the main build platform and the corresponding projection images. **f** Fabrication of the channel roof using the aux platform and the corresponding grayscale mask image. **g**, **h** In-situ transfer of the channel roof to the previously built part with bonding connection in the Z-direction via the second exposure and the corresponding mask image. **i** Fabrication of the rest of the part using the main build platform.

always receives more light energy than the roof's top using the bottom-up-projection-based configuration. Combining these three mechanisms ensures the split happens between the channel roof and the aux platform rather than the roof and the resin vat when the aux platform moves up in the Z-direction. After the aux platform returns to the original position, the main platform moves down to form an $\varepsilon$-mm gap with the channel roof (Fig. 2g).

(5)  The mask image shown in Fig. 2h is projected to fully cure the layer containing the channel roof and, therefore, in-situ transfer it to the previously built part on the main platform. The polymer-polymer interface (or the polymer-glass interface for the base layer) is much stronger than the polymer-FEP film interface (Fig. 2b). Therefore, the newly cured layer will combine with the previously built layers on the main platform during the release process. At this point, the target channel has been created, and its height is only determined by the linear motion system, which is more reliable and accurate than light dose modeling of liquid resin. The resin trapped

inside the channel remains to be unpolymerized as they receive no energy input.

(6)  Using the normal VPP process, the rest $n$-mm thick cube is printed by the main platform with layer thickness $l$ mm until the end. The channel roof with a thicker thickness ($l_r$) will exponentially cut down the light dose delivered into the channel portion when curing the following layers. The energy caused by curing the layers that are far from the channel is negligible. By controlling the roof layer thickness $l_r$ and the subsequent layer thickness $l$ according to our analytical model (see Supplementary Method III for modeling details), the accumulated light dose at the channel top is less than the critical dose $D_c$ as shown in the previous result (the bottom half of Fig. 1e and Supplementary Fig. 2b). Therefore, the over-curing issue can be effectively resolved in IsT-VPP.

If the channel height $h$ is equal to or smaller than the predefined $\varepsilon$, step 2 of the printing process (Fig. 2e) can be skipped. Meanwhile, in step 4 (Fig. 2g), let $\varepsilon = h$ so that the

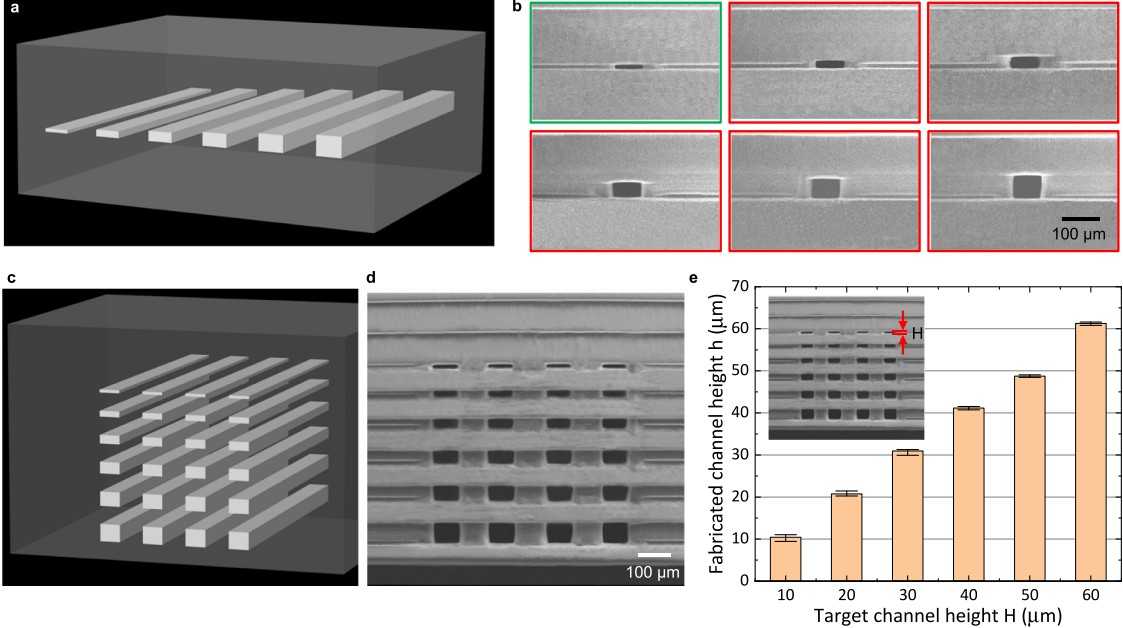

**Fig. 3 Fabrication of microfluidic channels via IsT-VPP. a** CAD model of the part with single-layer microfluidic channels embedded. The channel heights range from 10 μm to 60 μm. **b** SEM images of microfluidic channel cross-section. The channel circled by green has 10 μm in height. Such a channel is fabricated when connected with the previously built part corresponding to Fig. 2 g. The channels circled by red with heights of >10 μm are manufactured using the main build platform corresponding to Fig. 2e except for the last 10 μm layer. **c** CAD model of the part with multiple layer microfluidic channels. The channel heights range from 10 μm to 60 μm. **d** SEM image of the multi-layer microfluidic channel cross-section. **e** Statistical results of the fabricated channel heights.

channel with the designed height is fabricated when combining the roof and the built part. The IsT-VPP experimental prototype is shown in Supplementary Fig. 3. For multiple layers containing channel roofs in a CAD model, the flowchart of the printing process is given in Supplementary Fig. 4 and a printing demo can be found in Supplementary Movie 2. Suppose the roof layer contains the sidewalls of other vertical channels. Their surface quality will not be affected by the aux platform since the vertical channels can be printed using the main platform with the regular layer thickness (see Supplementary Fig. 5).

**Microfluidic channels with 10 μm height.** Microchannel resolution is a significant issue for all 3D printing processes[16]. Large 3D-printed internal microfluidic features with dimensions between 100 μm and 500 μm lack the capability of allowing the full range of microfluidic applications[12]. To verify the accurate microfluidic channel height control enabled by IsT-VPP, we first designed a test case containing one layer of embedded channels with 10, 20, 30, 40, 50, and 60 μm height (Fig. 3a). The channel width is 75 μm (4 pixels size of the used DLP system). The SEM image (Fig. 3b) gives a detailed cross-sectional profile of the printed channels. For the 10 μm-height channel (circled by green), since the channel height is equal to the gap size $\varepsilon = 10$ μm, the step shown in Fig. 2e was skipped, and the channel was fabricated when connecting the roof with the previously built layers (Fig. 2g). For the other channels (circled by red) with heights from 20 μm to 60 μm, partial channels were first fabricated using the main build platform (Fig. 2e). The fabrication result shows that all the channels are accurate with a good surface finish. To further verify the feasibility of the proposed method to fabricate multi-layer microchannels, we fabricated a model with 24 channels whose heights are from 10 μm to 60 μm (Fig. 3c). The SEM results (Fig. 3d) and the statistical result (Fig. 3e and Supplementary Table 2) show our method is accurate with an average error of <1.5 μm, which is

already the limit of the Z-linear stage used in the prototype system. The small error bars (<±1 μm) also indicate the reliability of our fabrication method. Some improvement suggestions to achieve higher channel height resolution are given in the Discussion session.

**Multifunctional automation components for microfluidic devices.** In addition to microfluidic channels, we next demonstrate IsT-VPP can benefit microfluidic automation components in microfluidic chips in terms of design flexibility, material options, and function extension. Current 3D-printed microfluidic valves either occupy a large space or have constraints in geometric design and material transparency. Using the IsT-VPP method, we can print transparent Quake-style microfluidic valves[40] with any gap size between the membrane and the bottom seat. This fabrication capability facilitates microfluidic valves design when integrated with channels of various heights that may enable novel applications.

The microfluidic valve consists of a flow channel allowing the liquid to pass through and a control channel overlaid orthogonally atop the flow channel (Fig. 4a). The control channel's vent is used for clearing uncured resin after printing and is sealed before use. The membrane-like suspended region (700 μm × 800 μm × 25 μm) between the control channel and the flow channel is the valve serving as the switch (Fig. 4b, c). The valve is open at rest so that blue-dyed deionized (DI) water can traverse (Fig. 4i). When applying sufficient pressure via the control channel, the membrane will deflect and contact the half-pipe ramp below (Fig. 4d, e). The maximum vertical gap from the ramp's top surface to the membrane's bottom surface is 40 μm (Fig. 4c). The DI water between the ramp and the membrane center is displaced, and the valve is closed (see the areas enclosed by yellow dash lines in Fig. 4i and Fig. 4j for a comparison). This process can be modeled as a rectangular plate with all the edges fixed under uniformly distributed

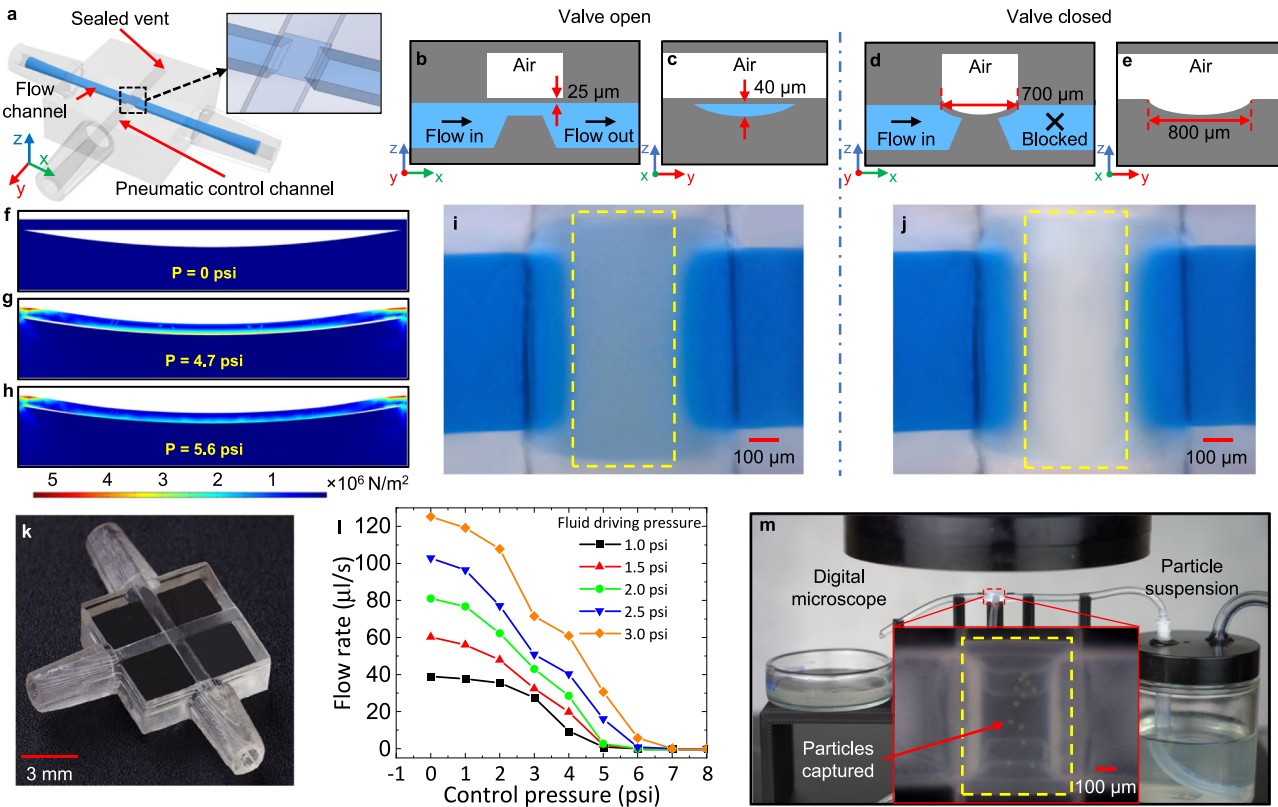

**Fig. 4 3D-printed microfluidic valve and specimen platform for automation. a** CAD model showing the configuration of the 3D-printed microfluidic valve. The vent at the end of the control channel will be sealed via construction glass glue after clearing uncured resin. The inset gives the detailed structure of the membrane and the ramp. Schematic diagram showing the front **b** and side **c** sectional view of the valve filled with blue-dyed DI water in the open state. Schematic diagram showing the front **d** and side **e** sectional view of the valve in the closed state. The liquid cannot pass through. The membrane is fully conformal to the curved ramp surface under pressure. FEA simulation of the valve membrane deflection at different pressures. **f** 0 psi. **g** 4.7 psi. **h** 5.6 psi. The color heat map shows the von Mises stress on the membrane at these pressures. **i** Microscope image showing the top view of the valve in the open state. **j** Microscope image showing the top view of the valve in the closed state. The liquid between the ramp and the membrane is displaced. **k** Fabrication result of the microfluidic valve (isometric view). **l** Closing pressure test of the fabricated valve under varying fluid driving pressures. **m** Specimen platform for particle samples observation and measurement. The deflected membrane captures the green particles.

loading from air pressure. The maximum deflection can be approximated by[47,48]

$$y_{max} = \frac{\alpha P b^4}{E t^3} \qquad (2)$$

where $P$ is the air pressure applied, $b$ and $t$ are the smaller sides and the thickness of the membrane, respectively, $E$ is Young's modulus, and $\alpha$ is the coefficient ($\alpha \approx 0.0138$ in our case).

For the used clear resin ($E$ is 164 MPa), Eq. 2 predicts the membrane will deflect by 40 μm at a pressure of 4.5 psi. Finite element analysis (FEA) predicts a 4.4% higher pressure required to achieve the deflection of 40 μm. The loading stress simulations indicate that the membrane (shown at rest in Fig. 4f) is maximally deformed at the center point and begins to contact the ramp at ~4.7 psi (Fig. 4g) and nearly conforms to the ramp's curved surface at ~5.6 psi (Fig. 4h). Figure 4k shows the fabrication result using IsT-VPP. We tested the valve's closing pressure by measuring the flow rate under different driving pressures applied to the flow channel (Supplementary Fig. 6). The flow channel inlet is connected to a bottle containing blue-dyed DI water. An electrically controlled air pressure source pressurizes the bottle to drive the DI water into the flow channel. The flow rate was measured by the mass of the collected DI water using a precision balance. A higher fluid driving pressure requires a correspondingly higher closing pressure to stop the flow (Fig. 4l). By applying pressure of 4 psi larger than the flow channel's driving

pressure, the flow rate approached ~ 0 μl/s but was not entirely stopped. Therefore, we set the closing pressure to 5 psi higher than the fluid driving pressure due to our setup resolution (e.g., 3 psi for fluid driving pressure and 8 psi for control pressure; Supplementary Movie 3). In this case, the valve can always ensure a successful closure. The test results agree well with the FEA simulation. The slightly light blue color that appears near the top and bottom of the ramp (see the area enclosed by yellow dash lines in Fig. 4j) shows residual liquid. This innocuous defect was caused by the staircase effect, the nature of 3D printing processes when fabricating a slope, as most 3D printing processes are essentially repeated layer-based 2D printing. The valve seat was designed as a ramp and was fabricated as a series of 2D layers. They can be seen in the inset of Fig. 4m (highlighted in yellow dash lines). A conservative estimation of the staircase effect on the valve performance was given in Supplementary Discussion, showing the influence can be ignored.

In addition to microfluidic valves, the 3D-printed fluid automation device can function as a specimen platform to replace costly robotic pipettors or tedious manual pipetting with the accurate control of the gap between the membrane and the ramp (Fig. 4m and Supplementary Movie 4). By placing the device with the membrane valve under a microscope, the flow channel works as a microscope slide; meanwhile, the membrane functions as a coverslip. Particle/cell samples suspensions are pressurized into the device. Once inflated, the membrane will

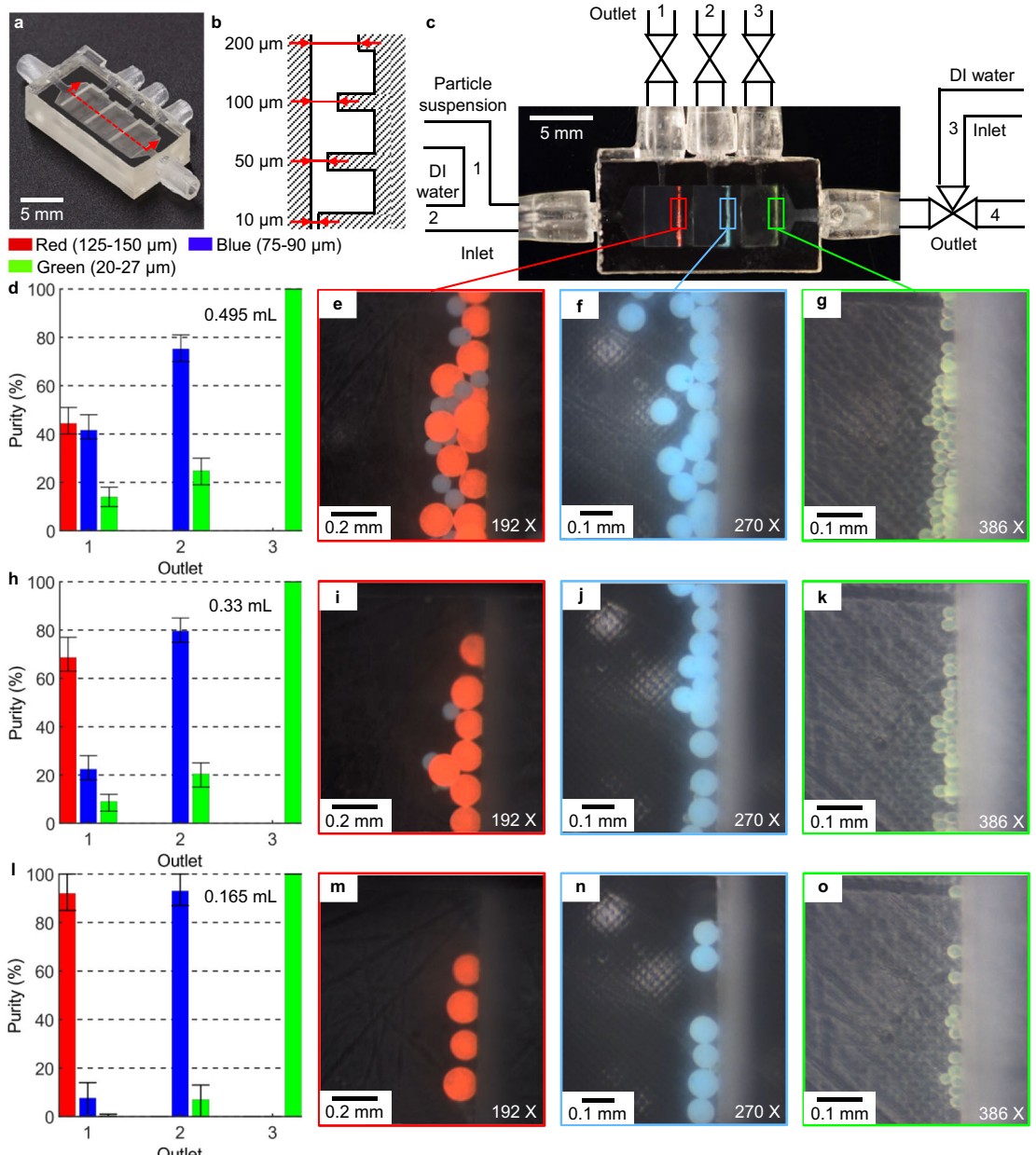

**Fig. 5 3D-printed microparticle sorting device. a** Fabrication result of the microfilter chip (isometric view). **b** Schematic drawing illustrating the interior of the microfilter chip (not to scale, side sectional view). **c** Illustration of the working principle of the microparticle sorting system (top view). **d–o** Particle sorting behaviors with different doses. **d**, **h**, **i** Purity of the particles from each outlet. **e–g**, **i–k**, **m–o** Microscope images showing the sorted microparticles in each microfilter reservoir.

deflect and capture the particle/cell samples for observation and measurement in real-time. The gap size (≤40 μm) ensures that only a single layer of particles/cells (20–27 μm) can pass through the microscope's field of view so that no overlap of particles/cells will occur.

**3D-printed microparticle sorting device.** Developments in cell-based clinical diagnosis and therapy have significantly increased the demand for techniques to purify micrometer-sized particles, including living cells[9]. Microfluidic systems have been recognized as simple, effective, and precise platforms for microfiltration. However, most 3D-printed interior channels are currently in the sub-millifluidic range (0.5 mm–1.0 mm), limiting many analytical applications. To demonstrate IsT-VPP's practicability, we

designed and built a weir-type microfilter system enabled by accurate channel gap control.

The isometric view of the 3D-printed microfilter chip and the interior structure's schematic are shown in Fig. 5a, b. The device's fundamental structure is a set of microchannels with decreased gap sizes in the cross-section working as a barrier in the flow direction (from 10 μm to 200 μm). The three weir-like, equally spaced barriers form gaps with the channel roof. Only particles smaller than the gap are permitted to pass over the obstruction, while larger particles will be captured and accommodated by the corresponding reservoir and collected later. The whole micro-particle sorting system is shown in Fig. 5c. The system consists of three inlets for particle suspension (inlet 1), DI water (inlet 2 and 3), and four outlets for red particles (outlet 1), blue particles (outlet 2), green particles (outlet 3), and DI water (outlet 4). The

working principle and sequences are shown in Supplementary Fig. 7 and Supplementary Movie 5. In phase 1, a certain particle mixture is introduced into the chip via inlet 1 with outlet 4 open. In phase 2, DI water flows in and out of the chip through inlet 2 and outlet 4, carrying particles to each corresponding reservoir. In phases 3–5, green, blue, and red particles are flushed by DI water coming from inlet 2 and 3 and recovered from outlets 3, 2, and 1 in order.

To validate the 3D-printed microfluidic device's effectiveness, we collected particles from outlets 1–3 and examined the purity of the particles from each outlet by measuring their relative ratios under different doses—volume processed in one cycle, as shown in Fig. 5d–o. The purity of green particles from outlet 3 can always reach 100% at any dose, which benefits from the precise channel height control and the sized-based approach (Fig. 5d, g, h, k, l, and o). We observed a dominant portion of blue particles (75.2%) in reservoir 2 and consequently drained from outlet 2 while with some green particles (24.8%) mixed in, which reduces the purity (Fig. 5d, f). The phenomenon is more evident in outlet 1 (Fig. 5e). The earlier-arrived large red particles will block the passway so that the following middle-size blue particles cannot squeeze through the gap into the next reservoir. For the small green particles, the gap is still big enough to pass through. So the ratio of green particles collected from outlet 1 is much less than blue ones (Fig. 5d). The particle sorting behaviors were similar if the dose decreased to 0.33 mL (Fig. 5h–k) except for more red particles and blue particles recovered from outlets 1 and 2, respectively. At the amount of 0.165 mL (Fig. 5l–o), particles purity from all three outlets can reach above 92% simultaneously, confirming the device can work as designed.

## Discussion

This work presents a VPP-based 3D printing methodology to accurately fabricate microfluidic channels with high $Z$-resolution (within 10 μm) and accuracy (±1 μm). An aux platform was integrated to manufacture channel roof, dramatically cutting down the energy penetration to the liquid resin in the channels. Thus, the constraint on minimum printable channel height due to the light penetration depth has been broken. More importantly, the channel height is determined by the $Z$-linear stage accuracy instead of the analytical model of light dose for photocurable resins. We envision this high $Z$-resolution printing ability will significantly boost the capabilities of 3D-printed microfluidic chips. More material options can be available for microfluidics using the VPP process. Material scientists could create new resins or fine-tune existing ones solely from users' needs, such as elasticity and biocompatibility, without considering its printability. The separation force study for different material interfaces validated the feasibility and reliability of the IsT-VPP method.

Multiple microfluidic devices were realized by software-generated mask images for the main and aux build platforms, including fluid routers, microvalves, and particle sorting chips. The USC-shaped fluid router demonstrated the ability to print 3D serpentine channels. With the aux platform, the transparent microvalve's gap can be <40 μm, breaking the design constraint and facilitating functional extension, miniaturization, and integration with different channel heights. The microfluidic specimen platform can further switch samples for convenient and automated measurement if integrated with multiple valves and sample suspension sources. A microfilter chip was fabricated with accurately controlled gap size, successfully realizing various microparticles sorting with high purity. SEM images, FEA simulation, and experimental videos verified the effectiveness of our fabrication results. The purity of the 3D-printed microparticle sorting device can be further enhanced by cascading multiple microfilters

or adding external forces such as acoustic force[49] to introduce shuffling at the barriers. Besides, the commonly used parallelization strategies can also be applied here to multiply the throughput[50,51]. Since the last gap size is 10 μm, all the particles <10 μm can also be sorted if needed. Notably, the aux platform working as the constrained surface also contributed to high channel surface quality, which could benefit applications that require smooth surfaces to prevent issues such as cell adhesion. Therefore, the presented IsT-VPP proves to be a valuable fabrication method that is applicable to general microfluidic particle/cell sorters and may facilitate the application of 3D-printed microfluidic systems in biological studies and clinical diagnosis. In addition to 3D-printed microfluidic devices, we envision this fabrication method of using double exposures and an additional build platform could open a new avenue for VPP development and other applications.

In future work, the performance and functionality of the IsT-VPP-printed microfluidic devices will be further enhanced and expanded. For instance, (i) as the resolution of a manufacturing system is determined by many aspects, including hardware and firmware, higher channel resolution like 1 μm could be achieved by utilizing a piezo-based $Z$-linear stage with sub-micron resolution, which is close to that of TPP. Note, even with a lower resolution (at 10 μm right now), the VPP process is over 100 times faster than the TPP process and has over 100 times lower cost; (ii) optical approaches[35,39,41] can be incorporated to handle transparent resins of much higher light penetration depth without compromising channel resolution; (iii) projectors with smaller pixel size can be applied to achieve truly micro-resolution in all $X/Y/Z$ directions; (iv) biocompatible transparent photocurable resins such as SprintRay's Surgical Guide 3 and EnvisionTec's E-Guard can be used to create microfluidic devices suitable for living cells; (v) the microvalves' size can be further scaled down using photocurable resins with a smaller Young's modulus; and (vi) large-scale parallelization and acoustic force can be performed for particle sorting devices for higher efficiency and purity. In summary, the presented IsT-VPP process has been demonstrated as a facile and general fabrication method for 3D-printed microfluidic channels with precisely controlled channel heights.

## Methods

**Materials**. Transparent photocurable resin (Anycubic Clear) was purchased from Anycubic Corporation. PDMS (SYLGARD[TM] 184 Silicone Elastomer Kit) was purchased from Dow Silicones Corporation. The FEP film was purchased from Shanghai Witlan Industry Co., Ltd and used as received. The pigments UVO[TM] (red and yellow) and Silc Pig[TM] (cyan) were purchased from Smooth-On, Inc. The blue coloring dye (alcohol ink set) was purchased from LET'S RESIN. The polyethylene microspheres with a diameter range of 20–27 μm (UVPMS-BG-1.025, red), 75–90 μm (UVPMS-BB-1.13, blue), 215–150 μm (UVPMS-BR-0.995, red) were purchased from Cospheric LLC and dispersed in DI water with the all-purpose cleaner (Simple Green) as a surfactant according to the procedure provided by the particle manufacturer. 99.5 % isopropanol (I-MAX) was purchased from Amazon.

**IsT-VPP fabrication and characterization of microfluidic chips**. First, the exposure time for different layer thicknesses was set based on the measured curing rate results and the fitted Jacob's working curve (Supplementary Method I and Fig. 1c). For a given layer thickness $l$, the curing depth is usually set in the range of 1.1~1.5 $l$ to ensure good bonding between adjacent layers and surface quality. Therefore, the exposure time was set to cure 1.25–1.5 layers. To have good adhesion of the resin on the build platform, we print the first four adhesive layers for any print job with a longer exposure time. We set the layer thickness to 10 μm for delicate channel portions and 100 μm for non-channel portions for efficiency. The printing parameters setting for all the demos are listed in Supplementary Table 3. Next, the self-developed program generated the mask images and corresponding G-Code (see Supplementary Method IV for algorithm details). Then an in-house-built software system read the images and G-Code and executed the whole manufacturing process, including sending command and image data sequences to the motion controllers and the DLP projector. The PC-end software system written in C++ language was developed with Visual Studio 2015. The

linear stages (404XR Series, Parker Hannifin) used in our setup have bidirectional repeatability of ±1.3 µm. The lead is 5 mm/round. We used the microcontroller and the motor driver (KFLOP and KSTEP, Dynomotion) with a 1/16 micro-stepping function. Since the stepper motors have 200 steps per round, the smallest distance that the stage can move is ~1.56 µm. The 405 nm UV DLP projector from a commercial 3D printer (MoonRay Model S100, SprintRay Inc.) was used. We further modified the optical system to shrink the printing area to 24 mm × 15 mm. The resolution of the DMD chip of the projector is 1280 × 800. All the 3D prints were fabricated on glass sheets (8476K121) purchased from McMaster-Carr.

The fabricated parts were sonicated in 99.5% isopropanol for 4 min immediately after printing. Since the unpolymerized resin remains trapped inside the channel, a syringe filled with isopropanol was used for flushing out the residual resin inside the devices' channels. Finally, the parts were sonicated in isopropanol and DI water for another 4 min, respectively. All the operations were done under low-light conditions.

The USC-shaped fluid router has barbed connectors designed and printed, so 1/16 in. hose (5239K24, McMaster-Carr) can be directly plugged in them. The cross-sectional channel profiles of the USC-shaped fluid router (Fig. 1f–h) and the straight channels (Fig. 3) were measured using a scanning electron microscope (JSM-7001F-LV, JEOL USA Inc.).

Before fabricating the microfluidic valve, the structural mechanics module of the FEA software (COMSOL 5.5 from COMSOL Inc.) was utilized to simulate the valve membrane deformation under different pressure loadings. The CAD models of the membrane and the half-pipe ramp were built in SolidWorks and then imported into COMSOL. The membrane's bottom surface and the ramp's top surface were treated as a contact pair for the boundary condition so that the ramp constrains the membrane deformation. The part was defined as a linear elastic material with the physical properties of Anycubic Clear. Different boundary loading pressures were applied to the membrane's top surface. In the physical experiment, pressure gauge (3846K433, McMaster-Carr), air pressure regulator (AW2000-02, Tailonz Pneumatic), and air compressor (BTFP02012, Stanley Bostitch) worked together to drive the liquid in the flow channel and provided air pressure to the control channel. Digital analytical balance (USS-DBS15-2, U.S. Solid Inc.) was utilized to measure the liquid flow rate, as shown in Supplementary Fig. 6. After the fluid flow stabilizes under a certain fluid driving pressure and control pressure, we collected water from the outlet for 1 min and measured its mass. The concentration of the particle suspension shown in Fig. 4m was ~$8 \times 10^3$ particles per mL. The suspension was continuously introduced into the specimen platform using a self-built syringe pump at a low float rate of 0.01 mL/min intentionally for easier video capturing by the digital microscope.

In the particle sorting experiment, the mixture of particles was first weighed and then suspended in an aqueous solution. The total particle concentration is ~$10^3$ per mL. The ratio of the particles is about 1:3:6 (red: blue: green) according to the table of the number of spheres per gram provided by the manufacturer. The particle suspension and DI water were introduced into the sorting device and collected sequentially by numerically controlled syringe pumps and electrical valves. To facilitate the digital microscope capture particles' movement, we set the flush rate as low as 0.05 mL/min in Supplementary Fig. 7 and Supplementary Movie 5. The distribution ratio of particles was defined as the absolute number of one kind of particles recovered from a specific outlet divided by the total number of particles retrieved from the same outlet. After a large quantity of particle suspension was collected from each outlet, we used droppers to sample the particles until we count 100 particles under a microscope and then do statistical analysis. Each measurement was repeated five times.

## Data availability

The data generated in this study are provided in the Supplementary Information and the figshare database with the link https://doi.org/10.6084/m9.figshare.17903738.v1[52]

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

## Acknowledgements

Y.C. acknowledges support from NSF through Grant CMMI-1151191, CM-1812675, and CMMI-1663663.

## Author contributions

Conceptualization: Y.C. Methodology: Y.X., H.M., Y.C. Software: Y.X., H.M. Validation: Y.X., F.Q., H.M. Formal analysis: Y.X., F.Q., S.L. Investigation: Y.X., F.Q., H.M., S.L., Y.Z., J.G., L.W. Writing—original draft: Y.X., H.M. Writing—review and editing: Y.C., N.M. Visualization: Y.X., F.Q., S.L. Supervision: Y.X., Y.C., N.M.

## Competing interests

Y.C., Y.X., and H.M. are holders of a provisional U.S. patent related to this work (filed in August 2021). The authors declare no competing interests.
