## [Peer Review File · Nature Communications]

REVIEWER COMMENTS

Reviewer #1 (Remarks to the Author):

Chen and coworkers report a new 3D printing approach that they named In-situ-Transfer vat photopolymerization (IsT-VPP) with special application in microfluidics. This innovative technology overcome critical issues in the 3D printing of microchannels due to the over-curing of the layers using standard vat photopolymerization techniques such as sterolithography or DLP. They demonstrated the fabrication of channels with 10 um height (Z-direction) using commercial resins and common light sources. This is big step for the microfluidics community due to the possibility of use commercial and transparent resin. Also, it avoids tedious fabrication steps using two different techniques. However, I have some concerns about its applicability in real-life:

1. The authors state that "Current commercial VPP machines can be easily modified to incorporate our method." However, I do not see so simple to incorporate an extra head - Aux platform in the paper- which is driven by two additional motorized stages. The commercially available printers are very often closed system and the space for adding this platform might be not there. Also, how can one control this platform and that communicate with the printed head of the existing system?

2. How universal is the methodology to generate the mask images for the called "second exposure"? How to implement it in a normal printing file?

Additional points to the authors:

1. Macrofluidic valve demonstrator (Figure 4j): there is some blue colour after closing the valve. Is this due to the residual liquid? Or this the valve not completely tight?

2. The authors state in the outlook that "biocompatible photocurable resins such as poly(ethylene glycol) diacrylate (PEGDA)-based resins can be used to create microfluidic devices suitable for living cells". I am afraid that for hydrogel-like structures, swelling effects will need to be taken into account and a much lower resolution will be achieved.

3. The discussion section is rather short. Some parts included in the results section might belong to the results section.

Reviewer #2 (Remarks to the Author):

Comments to the Authors

This manuscript reports a very innovative method for DLP stereolithography to realize much smaller negative space heights (channels, chambers, etc.) with 3D printing for microfluidics without increasing the resin optical absorption. In essence, the authors present a method that replaces resin optical absorption dependence of negative space height with the simplicity of mechanical positioning. Their approach takes clever advantage of the different adhesion forces of just-polymerized resin with different materials (PDMS, FEP, glass) to enable use of an auxiliary build platform to pre-structure polymerized resin on the bottom of the vat to protect negative space regions in the previous layer from subsequent light exposure while still ultimately incorporating the structured layer into the 3D print itself.

This work provides a new and highly attractive way of dramatically increasing the applicability of 3D printing for fluidic devices at the microfluidic size scale and as such is quite significant, and worthy of publication in Nature Communications. The authors have produced a clear and well-organized manuscript that is highly readable, with supplementary videos that are very helpful to understand their approach and operation of demonstration devices. I have a few suggestions for improvement below, after which I recommend that the paper be accepted for publication.

1. Some of the work in the manuscript does not acknowledge previously published work. For example, the demonstrated valves are the same design as the Folch group reported in Ref. 40. However, Ref. 40 is not referenced in this context. Likewise, the mathematical model for dose as a function of z in the "Light dose distribution" subsection is essentially the same as developed by the Nordin group in Refs. 35 & 39, but is not attributed to this previous work.

2. The valves are claimed to have a "half pipe ramp" shape, but the shape must clearly be stair-stepped because the valve seat is fabricated as a series of several layers. How does this affect operation of the valve and leakage in its closed state? For example, in Supplementary Video 3, the droplet at the end of the waste tube clearly grows when the valve is in a closed state. Also, the closed valve in Fig. 4j has light blue regions above and below the closed region in the photo, which indicates that at least 2 paths for fluid flow exist when the valve is closed.

3. The authors' new method relies on absorption of light in the specially fabricated "roof" layers by making the roof region thick compared to the optical absorption distance in the material. This leads to several questions:

- Is this an effective strategy if the photoinitiator absorption wavelength range does not fully cover the emission spectrum of the 3D printer LED? Clearly, if this is the case, the proposed method will not attenuate all of the source wavelengths, which would seem to seriously compromise the effectiveness of the method.

- Every "roof" layer is quite a bit thicker than regular layers. What effect will this have on the generality of what can be fabricated, specifically for microfluidics? For example, does this limit how

small the x-y cross section of a vertical channel can be that goes through a roof layer somewhere else in the 3D print?

4. A main stated motivation for the paper is to avoid UV absorbers that add some color to a fabricated device. What about UV absorbers that yield colorless devices, such as the biocompatible material reported in DOI: 10.1021/acsabm.0c00055?

5. The legend for "Accumulated" in Fig. 1e should be a solid line instead of a dashed line.

6. This is not a recommendation for an improvement, but, rather, to acknowledge some especially well done aspects of the paper:

- The figures are extremely clear and helpful. I really appreciate the care that went into creating such high quality figures.

- Supplementary Table 1 is excellent, as is its summary in Fig. 1a.

- In general, the Supplementary Information file is great, and adds a lot of value to the paper.

Reviewer #3 (Remarks to the Author):

Artykuł przedstawia bardzo interesujące rozwiązanie umożliwiające drukowanie mikrokanatów o minimalnej wysokości 10 mikrometrów. Jest to wysokość kanału nieosiągalna dla większości typowych drukarek stereo litograficznych (w tym DLP). Jedynie drukarki dwufotonowe mogą być w tym zakresie konkurencyjne. Artykuł jest napisany bardzo dobrze. Opis eksperymentu i wyniku jest jasny i dobrze udokumentowany. Mimo jednak ogromnego potencjału i niewątpliwej nowości przedstawionych wyników artykuł powinien być opublikowany w bardziej specjalistycznym czasopiśmie (na przykład ...).

Uwaga szczegółowa dotyczy kwestii minimalnej rzeczywistej wysokości mikrokanatu, którą można uzyskać opisaną metodą, z uwzględnieniem powtarzalności i możliwości wypłukania z mikrokanatu nieutwardzonego rezystu. Również wskazane byłoby porównanie uzyskanych wyników z konkurencyjną techniką druku jaką jest druk dwufotonowy. Być może autorzy wykazali by wówczas, że możliwe jest drukowanie mikrokanatu z submikrometryczną dokładnością i rozdzielczością z wykorzystaniem znacznie tańszej i prostszej techniki druku.

W konkluzji, mimo wielu zalet przedstawionego artykułu nie rekomenduję go do publikacji w Nature Communications.

The article presents a very interesting solution for printing microchannels with a minimum height of 10 micrometers. This channel height is unattainable by most common stereolithographic printers (including DLP). Only two-photon printers can be competitive in this respect. The article is written very well. The description of the experiment and the result is clear and well documented. However, despite the enormous potential and undoubtedly novelty of the presented results, the article should be published in a more specialized journal related to fabrication of microfluidic chips (for example LabChip Journal, Sensors and Actuator B:Chemical).

A detailed remark concerns the issue of the minimum real microchannel height that can be obtained by the described method, taking into account the repeatability and the possibility of washing out the uncured resist from the microchannel. It would also be advisable to compare the obtained results with a competing printing technique such as two-photon printing. Perhaps the authors would then demonstrate that it is possible to print a microchannel with sub-micrometric accuracy and resolution using a much cheaper and simpler printing technique (as a groundbreaking result worth to be published in Nature Communications).

In conclusion, despite many advantages of the presented article, I do not recommend it for publication in Nature Communications.

Dear Reviewers:

Thanks for your decision letter and for the reviewers' comments concerning our manuscript entitled "In-situ-Transfer Vat Photopolymerization for Transparent Microfluidic Device Fabrication". These comments and suggestions are all valuable and very helpful for revising and improving our paper, as well as providing the important guiding significance to our research. We have carefully studied the comments and tried our best to make the revision of the paper that we hope to meet with approval. The main corrections are marked red in the revised manuscript, and the responses to the editor's and reviewers' comments are listed as follows:

Reviewer #1:

Chen and coworkers report a new 3D printing approach that they named In-situ-Transfer vat photopolymerization (IsT-VPP) with special application in microfluidics. This innovative technology overcome critical issues in the 3D printing of microchannels due to the over-curing of the layers using standard vat photopolymerization techniques such as sterolithography or DLP. They demonstrated the fabrication of channels with 10 um height (Z-direction) using commercial resins and common light sources. This is big step for the microfluidics community due to the possibility of use commercial and transparent resin. Also, it avoids tedious fabrication steps using two different techniques. However, I have some concerns about its applicability in real-life:

1. The authors state that “Current commercial VPP machines can be easily modified to incorporate our method.” However, I do not see so simple to incorporate an extra head - Aux platform in the paper- which is driven by two additional motorized stages. The commercially available printers are very often closed system and the space for adding this platform might be not there. Also, how can one control this platform and that communicate with the printed head of the existing system?

Response:

Thanks for the question, and sorry for the confusion. We have rephrased the statement as follows: “Current commercial and research VPP developers can easily apply our method to their systems” (see page 6 of the manuscript). We also added a section in the supplementary information to give researchers and technicians some basic guidelines for extending their DLP/LCD 3D printers (Supplementary Information V. Guidelines of IsT-VPP setup based on common DLP/LCD 3D printers, page 24 of the supplementary information).

2. How universal is the methodology to generate the mask images for the called “second exposure”?
How to implement it in a normal printing file?

Response:

Thanks for the question. The mask images for the first and second exposure are generated by Boolean operation of the channel and channel-roof layer’s mask images. The mask images, namely matrices, can be manipulated in programming software like C++ or Matlab (see Supplementary Information IV. Mask image planning, page 19 of the supplementary information). As to how to implement it in a normal printing file, unfortunately, we do not know what such a printing file looks like since we built the IsT-VPP system from scratch by ourselves (refer to Supplementary Figure 3) without using a commercial printer. In general, we think the software to generate the printing file needs to be revised, if it is open-sourced, to incorporate the additional hardware and the presented building process; if the software is not open-sourced, the user may need to understand the syntax of the printing file so an additional micro-controller can be integrated with the commercial printer to incorporate the added linear stages and the presented building process (see Supplementary Information V. Guidelines of IsT-VPP setup based on common DLP/LCD 3D printers, page 25 of the supplementary information).

3. Microfluidic valve demonstrator (Figure 4j): there is some blue colour after closing the valve.
Is this due to the residual liquid? Or this the valve not completely tight?

Response:

Thanks for the question. The slightly light blue color that appears near the top and bottom of the ramp (see the area enclosed by yellow dash lines in Fig. 4j) shows residual liquid. This innocuous defect was caused by the staircase effect, the nature of 3D printing processes when fabricating a

slope, as most 3D printing processes are essentially repeated layer-based 2D printing. The valve seat was designed as a ramp and was fabricated as a series of 2D layers. They can be seen clearly in the inset of Fig. 4m (highlighted in yellow dash lines). A conservative estimation of the staircase effect on the valve performance was given in Supplementary Discussion (see page 25 of the supplementary information), showing the influence can be ignored.

Fig. 4 3D-printed microfluidic valve and specimen platform for automation.

Due to the staircase effect, there will be residual liquid after closing the valve in the gap between the valve membrane and the valve seat at the region highlighted by yellow dash lines in the inset of Fig. 4m. However, based on the design, this gap should be smaller than the layer thickness (10 μm used in our case) – suppose the average gap is 5 μm. We can estimate the gap effect based on

the Darcy–Weisbach Equation $\frac{\Delta p}{L} = \frac{128 \mu Q}{\pi D^4}$ given the following parameters (see the table below).

The flow rate Q will be as low as 118×10^{-5} $\mu\text{l/s}$, which can be ignored. This is a conservative estimation since we have not considered the pressure drop caused by the length of the channel and the plastic hose. Hence, even there is residual liquid in the gap after closing the valve, the residual liquid, based on the design, will flow very slowly without affecting the valve's performance. We added the above description in the manuscript (see page 18-19 of the manuscript) and supplementary information (see Supplementary Discussion, page 25 of the supplementary information).

Pressure drop Δp	3 psi
Valve seat length L	300 μm
Dynamic viscosity of water μ	8.90×10^{-4} Pa·s
Hydraulic diameter D	5 μm

In the experiment, we used a pressure gauge (3846K433, McMaster-Carr), an analog air pressure regulator (AW2000-02, Tailonz Pneumatic), and an air compressor (BTFP02012, Stanley Bostitch) to provide the air pressure to the control channel. As the pressure in the control channel increases, the flow rate will decrease. By applying the pressure of 4 psi larger than the flow channel's driving pressure, the flow rate approached ~ 0 $\mu\text{l/s}$ but was not entirely stopped (as shown in the old Supplementary Movie 3). If we set the closing pressure of 5 psi higher than the fluid driving pressure, the valve will behave as the designed scenario. It can always ensure a complete closure, as shown in the new Supplementary Movie 3. We have updated Supplementary Movie 3 and Fig. 4j in the manuscript using the new setting. We added the description about the physical experiment in the manuscript (see IsT-VPP fabrication and characterization of microfluidic chips in the

Methods section, page 27 of the manuscript and Multifunctional automation components for microfluidic devices in the Results section, page 18 of the manuscript).

4. The authors state in the outlook that “biocompatible photocurable resins such as poly(ethylene glycol) diacrylate (PEGDA)-based resins can be used to create microfluidic devices suitable for living cells”. I am afraid that for hydrogel-like structures, swelling effects will need to be taken into account and a much lower resolution will be achieved.

Response:

Thanks for the comment. We have replaced the “poly (ethylene glycol) diacrylate (PEGDA)-based resins” with other biocompatible transparent resins that have been widely used for dental applications, such as SprintRay’s Surgical Guide 3 and EnvisionTec’s E-Guard (see page 24 of the manuscript). These resins have no swelling effect.

5. The discussion section is rather short. Some parts included in the results section might belong to the results section.

Response:

Thanks for the suggestion. We have moved part of the results section to the discussion section (see pages 22-24 of the manuscript).

Reviewer #2:

This manuscript reports a very innovative method for DLP stereolithography to realize much smaller negative space heights (channels, chambers, etc.) with 3D printing for microfluidics without increasing the resin optical absorption. In essence, the authors present a method that replaces resin optical absorption dependence of negative space height with the simplicity of mechanical positioning. Their approach takes clever advantage of the different adhesion forces of just-polymerized resin with different materials (PDMS, FEP, glass) to enable use of an auxiliary build platform to pre-structure polymerized resin on the bottom of the vat to protect negative space regions in the previous layer from subsequent light exposure while still ultimately incorporating the structured layer into the 3D print itself.

This work provides a new and highly attractive way of dramatically increasing the applicability of 3D printing for fluidic devices at the microfluidic size scale and as such is quite significant, and worthy of publication in Nature Communications. The authors have produced a clear and well-organized manuscript that is highly readable, with supplementary videos that are very helpful to understand their approach and operation of demonstration devices. I have a few suggestions for improvement below, after which I recommend that the paper be accepted for publication.

1. Some of the work in the manuscript does not acknowledge previously published work. For example, the demonstrated valves are the same design as the Folch group reported in Ref. 40. However, Ref. 40 is not referenced in this context. Likewise, the mathematical model for dose as a function of z in the “Light dose distribution” subsection is essentially the same as developed by the Nordin group in Refs. 35 & 39, but is not attributed to this previous work.

Response:

Thanks for the comment. We have added the references in the related context (see pages 7 and 16 of the manuscript).

2. The valves are claimed to have a “half pipe ramp” shape, but the shape must clearly be stair-stepped because the valve seat is fabricated as a series of several layers. How does this affect operation of the valve and leakage in its closed state? For example, in Supplementary Video 3, the droplet at the end of the waste tube clearly grows when the valve is in a closed state. Also, the closed valve in Fig. 4j has light blue regions above and below the closed region in the photo, which indicates that at least 2 paths for fluid flow exist when the valve is closed.

Response:

Thanks for the question. The slightly light blue color that appears near the top and bottom of the ramp (see the area enclosed by yellow dash lines in Fig. 4j) shows residual liquid. This innocuous defect was caused by the staircase effect, the nature of 3D printing processes when fabricating a slope, as most 3D printing processes are essentially repeated layer-based 2D printing. The valve seat was designed as a ramp and was fabricated as a series of 2D layers. They can be seen clearly in the inset of Fig. 4m (highlighted in yellow dash lines). A conservative estimation of the staircase effect on the valve performance was given in Supplementary Discussion (see page 25 of the supplementary information), showing the influence can be ignored.

Fig. 4 3D-printed microfluidic valve and specimen platform for automation.

Due to the staircase effect, there will be residual liquid after closing the valve in the gap between the valve membrane and the valve seat at the region highlighted by yellow dash lines in the inset of Fig. 4m. However, based on the design, this gap should be smaller than the layer thickness (10 μm used in our case) – suppose the average gap is 5 μm . We can estimate the gap effect based on

the Darcy–Weisbach Equation $\frac{\Delta p}{L} = \frac{128}{\pi} \frac{\mu Q}{D^4}$ given the following parameters (see the table below).

The flow rate Q will be as low as $118 \times 10^{-5} \mu\text{l/s}$, which can be ignored. This is a conservative estimation since we have not considered the pressure drop caused by the length of the channel and the plastic hose. We added the description in the manuscript (see page 18-19 of the manuscript) and supplementary information (see Supplementary Discussion, page 25 of the supplementary information).

Pressure drop Δp	3 psi
Valve seat length L	300 μm
Dynamic viscosity of water μ	$8.90 \times 10^{-4} \text{ Pa}\cdot\text{s}$
Hydraulic diameter D	5 μm

In the experiment, we used a pressure gauge (3846K433, McMaster-Carr), an analog air pressure regulator (AW2000-02, Tailonz Pneumatic), and an air compressor (BTFP02012, Stanley Bostitch) to provide the air pressure to the control channel. As the pressure in the control channel increases, the flow rate will decrease. By applying the pressure of 4 psi larger than the flow channel's driving pressure, the flow rate approached $\sim 0 \mu\text{l/s}$ but was not entirely stopped (as shown in the old Supplementary Movie 3). If we set the closing pressure of 5 psi higher than the fluid driving pressure, the valve will behave as the designed scenario. It can always ensure a complete closure, as shown in the new Supplementary Movie 3. We have updated Supplementary Movie 3 and Fig. 4j in the manuscript using the new setting. We added the description about the physical experiment in the manuscript (see IsT-VPP fabrication and characterization of microfluidic chips in the Methods section, page 27 of the manuscript and Multifunctional automation components for microfluidic devices in the Results section, page 18 of the manuscript).

3. The authors' new method relies on absorption of light in the specially fabricated "roof" layers by making the roof region thick compared to the optical absorption distance in the material. This leads to several questions:

- Is this an effective strategy if the photoinitiator absorption wavelength range does not fully cover the emission spectrum of the 3D printer LED? Clearly, if this is the case, the proposed method will not attenuate all of the source wavelengths, which would seem to seriously compromise the effectiveness of the method.

- Every “roof” layer is quite a bit thicker than regular layers. What effect will this have on the generality of what can be fabricated, specifically for microfluidics? For example, does this limit how small the x-y cross section of a vertical channel can be that goes through a roof layer somewhere else in the 3D print?

Response:

Thanks for the questions.

(1) If the photo initiator’s absorption spectrum does not fully cover the light source’s emission spectrum, the light energy outside the absorption spectrum will not affect the polymerization process. Also, for most commercial resins, the photo initiators’ absorption spectrum is designed to match the light source’s wavelength (e.g., 405 nm or 385 nm) to ensure the photopolymerization process initiates and propagates effectively and efficiently.

(2) If the roof layer contains the sidewalls of other vertical channels, the vertical channels can be printed using the main platform (and only the main platform) with the same layer thickness in step 2 (see Supplementary Fig. 5). Note the “channel roof” mentioned in the paper refers to the part of a layer that encloses the channel, not the whole layer (see line 119 on page 5 of the manuscript). Hence the thickening of the channel roof will not affect the surface quality or X-Y resolution of other vertical channels. We have added the description in the manuscript (page 14 of the

manuscript). We have added a figure to illustrate this process (see Supplementary Fig. 5 – also as follows).

Supplementary Figure 5 | Printing process illustration for a part with both horizontal and vertical channels. (a) The cross-section view of the sample part. (b-g) Fabrication process. Note the sidewalls of the vertical channel can be fabricated using the regular layer thickness in step (c) without being affected by the Aux platform in step (d) or (e).

4. A main stated motivation for the paper is to avoid UV absorbers that add some color to a fabricated device. What about UV absorbers that yield colorless devices, such as the biocompatible material reported in DOI: 10.1021/acsabm.0c00055?

Response:

Thanks for the question. The referred paper presented an effective method that tried to solve the over-curing issue from the material perspective. We have added it in the reference (see page 5 of the manuscript). Different from it, we used an approach that does not require careful calibration and light intensity control. In the revised manuscript, we have added statistical results of the fabricated channel heights (refer to Fig. 3e). It shows that the fabricated channel heights have small

error bars (less than $\pm 1 \mu\text{m}$). We believe such accuracy and reliability control are difficult for a material-based approach by adding UV absorbers.

5. The legend for “Accumulated” in Fig. 1e should be a solid line instead of a dashed line.

Response:

Thanks for the comment. We have updated the legend of Fig. 1e (see page 7 of the manuscript).

6. This is not a recommendation for an improvement, but, rather, to acknowledge some especially well done aspects of the paper:

- The figures are extremely clear and helpful. I really appreciate the care that went into creating such high quality figures.

- Supplementary Table 1 is excellent, as is its summary in Fig. 1a.

- In general, the Supplementary Information file is great, and adds a lot of value to the paper.

Response:

Thanks for the comment. We sincerely thank the reviewer for the suggestions on improving the paper's quality.

Reviewer #3:

The article presents a very interesting solution for printing microchannels with a minimum height of 10 micrometers. This channel height is unattainable by most common stereolithographic printers (including DLP). Only two-photon printers can be competitive in this respect. The article is written very well. The description of the experiment and the result is clear and well documented.

1. However, despite the enormous potential and undoubtedly novelty of the presented results, the article should be published in a more specialized journal related to fabrication of microfluidic chips (for example LabChip Journal, Sensors and Actuator B:Chemical).

Response:

Thanks for the comment. This paper presented a general fabrication method to 3D print features with micro gaps. Such types of features are ubiquitous and general in all kinds of engineering applications. We picked microfluidic chips as the test cases to demonstrate its fabrication capability; however, *we believe this fabrication method of using double exposures and an additional build platform benefits many other research areas beyond microfluidic devices fabrication*, such as surface chemistry, microsensor design, drug delivery, energy storage, and bio fabrication. Hence we decided to submit our work to a general journal such as “Nature Communications”. We hope our work can inspire researchers from multiple disciplines to benefit from our work, not limited to microfluidic chips. We have added the description of our fabrication method for other applications in the Discussion section (see page 24 of the manuscript).

2. A detailed remark concerns the issue of the minimum real microchannel height that can be obtained by the described method, taking into account the repeatability and the possibility of

washing out the uncured resist from the microchannel. It would also be advisable to compare the obtained results with a competing printing technique such as two-photon printing. Perhaps the authors would then demonstrate that it is possible to print a microchannel with sub-micrometric accuracy and resolution using a much cheaper and simpler printing technique (as a groundbreaking result worth to be published in Nature Communications). In conclusion, despite many advantages of the presented article, I do not recommend it for publication in Nature Communications.

Response:

Thanks for the comment. A novelty of our process is the controlled gap size that is now mainly determined by the z-linear stage accuracy. In our setup, the motorized z-linear stage (404XR Series, Parker Hannifin) has bidirectional repeatability of $\pm 1.3 \mu\text{m}$. The lead is 5 mm/round. We use the microcontroller and the motor driver (KFLOP and KSTEP, Dynamotion) with a 1/16 microstepping function. Since the stepper motors have 200 steps per round, the smallest distance the stage can move is $5/(200 \cdot 16) = 1.5625 \mu\text{m}$. We added the hardware information in the Methods section (see page 26 of the manuscript). Of course, the resolution of a manufacturing system is determined by many other aspects, including the process, hardware, and firmware. In the future, we plan to achieve a higher Z resolution by utilizing a piezo-based Z-linear stage with a sub-micron resolution, so it would be close to the resolution of the two-photon printing (TPP). Note, even with a lower resolution (at 10 μm right now), the VPP process is over 100 times faster than the TPP process and has over 100 times lower cost. We have added the description in the Discussion section (see page 24 of the manuscript).

As for washing, we followed the same process given by others' work, including the microfluidic devices fabricated by TPP or DLP with UV absorbers. The fabricated parts were sonicated in 99.5% isopropanol for 4 min immediately after printing. Since the unpolymerized resin remains trapped

inside the channel, a syringe filled with isopropanol was used for flushing out the residual resin inside the devices' channels. Finally, the parts were sonicated in isopropanol and DI water for another 4 min, respectively. All the operations were done under low-light conditions (see page 26 of the manuscript). Although the gap features are small in our work, the neighboring solid films are rigid so they can go through the washing processes without an issue.

REVIEWERS' COMMENTS

Reviewer #1 (Remarks to the Author):

The authors have addressed all my comments and I recommend for publication.

Reviewer #2 (Remarks to the Author):

The authors have satisfactorily addressed my comments as well as those of the other reviewers. I recommend publication in Nature Communications.